# Enhancing Prostate Cancer Staging: Association of 68Ga-PSMA PET/CT Imaging with Histopathological Grading in Treatment-Naive Patients

**DOI:** 10.3390/cancers16203526

**Published:** 2024-10-18

**Authors:** Oleksii Pisotskyi, Piotr Petrasz, Piotr Zorga, Marcin Gałęski, Pawel Szponar, Katarzyna Brzeźniakiewicz-Janus, Tomasz Drewa, Krzysztof Kaczmarek, Michał Cezary Czarnogórski, Jan Adamowicz

**Affiliations:** 1Department of Urology, Voivodeship Hospital Gorzów Wielkopolski, 66-400 Gorzów Wielkopolski, Poland; marcin.galeski@szpital.gorzow.pl (M.G.); pawel.szponar@szpital.gorzow.pl (P.S.); 2Clinical Department of Nuclear Medicine, University of Zielona Góra, 65-002 Zielona Góra, Poland; piotr.zorga@szpital.gorzow.pl; 3Clinic of Hematology, Clinical Oncology, and Radiotherapy with the Chemotherapy Subunit, University of Zielona Góra, 65-002 Zielona Góra, Poland; katarzyna.brzezniakiewicz-janus@szpital.gorzow.pl; 4Chair of Urology and Andrology, Ludwik Rydygier Collegium Medicum in Bydgoszcz, Nicolaus Copernicus University in Toruń, 85-790 Bydgoszcz, Poland; tomaszdrewa@cm.umk.pl (T.D.); jan.adamowicz@cm.umk.pl (J.A.); 5Department of Gynecology and Obstetrics, Voivodeship Hospital Gorzów Wielkopolski, 66-400 Gorzów Wielkopolski, Poland; krzysztof.kaczmarek@szpital.gorzow.pl; 6Ludwik Rydygier Collegium Medicum, Bydgoszcz Nicolaus Copernicus University, 85-790 Bydgoszcz, Poland; mcczarnogorski@gumed.edu.pl

**Keywords:** 68Ga-PSMA PET/CT, prostate cancer staging, histopathological grading, SUVmax correlation, biopsy vs. final pathology, PSMA expression

## Abstract

This study investigates the correlation between 68Ga-PSMA PET/CT imaging and histopathological grading in treatment-naive prostate cancer patients. We retrospectively analyzed 86 patients who underwent 68Ga-PSMA PET/CT scans for prostate cancer staging, with a subset of 40 patients also undergoing radical prostatectomy. The PET/CT results, specifically SUVmax values, were compared with Gleason scores and PSA levels to assess their association with cancer grades. Results indicated a significant positive correlation between SUVmax and ISUP grades, suggesting that higher SUVmax values correspond to more advanced prostate cancer stages. Furthermore, the analysis of biopsy and radical prostatectomy pathology reports revealed a substantial rate of grading changes, underscoring the limitations of biopsy alone for accurate staging. These findings highlight the potential utility of 68Ga-PSMA PET/CT as a diagnostic tool in stratifying prostate cancer risk and guiding treatment decisions. Future studies are recommended to validate these observations and explore further applications of this imaging modality.

## 1. Introduction

Prostate cancer (PC) ranks as the second most commonly diagnosed cancer in men, with incidence rates that vary globally [1]. Prostate-specific membrane antigen (PSMA), a type II transmembrane glycoprotein (also known as glutamate carboxypeptidase II), is physiologically expressed in tissues such as the salivary and lacrimal glands, normal prostate epithelium, sympathetic ganglia, duodenum, colon, and the proximal renal tubules [2].

PSMA-PET/CT is an advanced imaging modality that targets prostate cancer cells. The antigen, highly expressed in primary and metastatic prostate cancer, is also detected in various other tumors and their metastases due to neovascularization. Interestingly, uptake can occur in several benign granulomatous and inflammatory diseases as well [3]. The expression levels in prostate cancer are elevated by approximately 100–1000 times compared to normal tissue, though the precise mechanism behind this overexpression is not fully understood [4].

Prostate cancer is one of the tumors in which it is key to work in a multidisciplinary team in order to obtain the best possible results for patients [5]. To plan the most effective treatment, it is crucial to thoroughly evaluate both the extent of local involvement and systemic progression in newly diagnosed prostate cancer (PCa). Stratifying patient risk depends on multiple pre-treatment factors, including digital rectal examination (DRE) findings, prostate-specific antigen (PSA) levels, and the Gleason score (GS) obtained from prostate biopsy results [6]. Current guidelines recommend using MRI, computed tomography (CT), and bone scintigraphy for PCa staging to inform treatment decisions. Conventional imaging modalities, such as BS and CT, exhibit limited accuracy in identifying metastases to lymph nodes and bones, particularly among patients with low PSA levels. Consequently, the limitations of these imaging tools can lead to inaccurate assessment, potentially resulting in under-staging and under-treatment [7,8].

In the last 5 years, Ga-68 PSMA-PET/CT has become a revolutionary imaging technique for detecting PC relapse. Numerous studies have consistently illustrated that PSMA exhibits superior sensitivity and specificity compared to traditional approaches or choline PET, particularly in identifying tumor recurrence, especially in patients with low PSA levels (<1.0 ng/mL) [9]. While promising results suggest a significant clinical impact in altering the therapeutic approach, based on PSMA PET/CT evaluation of BCR, demonstrating improvement in long-term outcomes is crucial to validate the clinical utility of this transformative molecular imaging technique [10].

The aim of the present study was to assess the intensity of 68Ga-PSMA uptake in the primary tumor in patients with biopsy-proven treatment-naive PC, and to determine whether a correlation exists between 68Ga-PSMA accumulation in the primary tumor, and the Gleason score and/or PSA concentration. Moreover, we compared preoperative histopathology reports with definitive pathological reports after robot-assisted radical prostatectomy (RARP).

## 2. Material and Methods

### 2.1. Patients

We conducted a retrospective review of 86 patient records, all of whom underwent 68Ga-PSMA PET/CT imaging for prostate cancer (PCa) staging. Among them, 40 patients proceeded with robot-assisted radical prostatectomy (RARP). PCa was initially diagnosed for all patients via transrectal ultrasound-guided biopsy. Each patient had both an initial biopsy and a final analysis of postoperative specimens. In addition to using the Gleason scoring system, we categorized tumors according to the 2014 International Society of Urological Pathology Consensus (adopted by the WHO in the 2016 edition of Pathology and Genetics), as follows: grade group 1 (GS ≤ 6), grade group 2 (GS 3 + 4 = 7), grade group 3 (GS 4 + 3 = 7), grade group 4 (GS 4 + 4 = 8, 3 + 5 = 8, 5 + 3 = 8), and grade group 5 (GS 9–10).

### 2.2. Image Acquisition

Following the radiotracer’s preparation and quality assessment, each patient was administered an activity dose of 2 MBq per kilogram of body weight of 68Ga-PSMA, calibrated according to the radio-labeling yield. After the injection of the radiotracer, patients received 20 mg of Furosemide.

Whole-body imaging was performed 50 min post-injection of the radiotracer using one of two integrated PET/CT scanners: Discovery PET-CT 680 (Gorzów Wielkopolski, Poland). Patients were positioned supine on the scanner table, and a CT transmission scan was conducted without intravenous contrast enhancement. Following this, PET emission scanning was performed with a duration of 3 min per bed position, covering the same transverse field of view in the caudocranial direction. CT transmission images were used for attenuation correction, and image reconstruction was carried out using an iterative method. SUV values are related to lean body mass.

### 2.3. Image Analysis

PET images were reviewed using Advantage Workstation Server version 3.2 ext 1.2. Regions of interest (ROIs) were manually outlined around the prostate gland, ensuring that bladder activity was excluded to accurately determine the maximum SUV (SUVmax) values. A segmental analysis of the prostate was not performed. The highest SUVmax value observed within the entire prostate gland was recorded, representing the area with the greatest PSMA expression, for further analysis.

### 2.4. Statistical Analysis

All results were presented as median values and mean ± standard deviation. To evaluate correlations between various parameters, Pearson’s correlation test was applied. Grade group comparisons were conducted using the chi-square test. Receiver operating characteristic (ROC) curve analysis was employed to calculate sensitivity and specificity. All statistical tests for significance were two-tailed, with a *p* value of less than 0.05 indicating significance.

## 3. Results

A total of 86 male patients were included, with a median age of 68 years (range 42 to 82 years) (Table 1). Examples of 68Ga-PSMA PET/CT fusion images corresponding to each grade group are shown in Figure 1. The median PSA concentration was 11.3 ng/mL (range 2 to 250 ng/mL). No significant correlation was found between PSA values and tumor grade groups (Pearson’s ρ = 0.24, *p* = 0.027). However, there was a moderately positive correlation between SUVmax and tumor grade group (Pearson’s ρ = 0.34, *p* < 0.001) (Figure 1).

The Gleason score among patients ranged from 6 to 10. Patients presented with a median PSA concentration of 11.3 ng/mL. The median SUVmax among all tumors was 9.33 (range: 2.20–94.72).

There was a moderately positive correlation (Pearson’s ρ = 0.42, *p* < 0.001) between PSA concentration and SUVmax (Figure 2).

Patients with PSA levels exceeding 10 ng/mL showed higher 68Ga PSMA uptake in primary tumors compared to those with PSA levels below 10 ng/mL, with median SUVmax values of 12.98 and 6.09, respectively, demonstrating a statistically significant difference (*p* < 0.001).

Combining GS and tumor-related tracer (PSMA) uptake revealed an upward trend in elevation of the SUVmax according to a higher GS/ISUP grade: GS 6/ISUP I (SUVmax = 5.49), GS 7a/ISUP II (SUVmax = 7.73), GS 7b /ISUP III (SUVmax = 13.01), GS 8/ISUP IV (SUVmax = 15.74), and GS 9/ISUP 5 (SUVmax = 17.27) (Table 2). The same trend was found between the SUVmax and ISUP grade.

These tables illustrate the SUVmax values across different Gleason score (GS) and ISUP grade subgroups. An upward trend is observed in SUVmax values as the Gleason score and ISUP grade increase, indicating a relationship between higher tumor grades and increased PSMA tracer uptake. Specifically, the median SUVmax rises from 5.49 in GS 6/ISUP I to 17.27 in GS 9/ISUP V. This trend suggests that more aggressive tumors, as represented by higher GS and ISUP grades, demonstrate greater tracer uptake, reflecting increased metabolic activity associated with advanced disease.

Among the 86 patients, 40 underwent robot-assisted radical prostatectomy (RARP). Based on preoperative biopsy reports, patients were distributed across grade groups as follows: 13 (32.5%) in grade group 1, 8 (20%) in grade group 2, 9 (22.5%) in grade group 3, 6 (15%) in grade group 4, and 4 (10%) in grade group 5. Post-surgery, we observed 35% of cases upgraded and 22.5% downgraded in ISUP grade following radical prostatectomy. Comparing the prostatectomy pathology results with initial biopsy findings revealed grade group shifts in 57.5% of cases, a statistically significant difference (*p* < 0.001). Specifically, 7 patients experienced a one-level upgrade, 6 patients were upgraded by two levels, and one patient by three levels. Downgrades included a one-level decrease in 5 patients and a two-level decrease in 4 patients (Table 3).

After radical prostatectomy, 27.5% (*n* = 11) of patients transitioned from a low-risk category (grade groups 1 and 2) to a high-risk category (grade groups 3, 4, and 5), while 7.5% (*n* = 3) of patients shifted from a high-risk category to a low-risk one. An SUVmax cut-off of 5.64 was seen in 4 out of 11 patients (36.36%) who transitioned to higher-risk groups, indicating a potential pattern rather than a predictive threshold (Figure 3).

Additionally, SUVmax values for patients who underwent radical prostatectomy were compared with the final postoperative histological findings (*n* = 40). This comparison showed no statistically significant correlation (*p* = 0.289) between the median SUVmax and Gleason score (GS). Specifically, the group with a GS of 6 or 7 had a median SUVmax of 6.51 (range: 2.2–50.15), while GS 8–9 tumors exhibited a median SUVmax of 14.19 (range: 6.69–45.84).

Receiver operating characteristic (ROC) curve analysis of SUVmax (area under the curve = 0.82, 95% CI: 0.70–0.94) identified a cut-off value of 5.64, providing sensitivity and specificity rates of 76% and 60%, respectively.

## 4. Discussion

The primary objective of our retrospective study was to assess whether a correlation exists between the intensity of 68Ga PSMA uptake in primary prostate cancer patients and their Gleason score (GS) and/or ISUP grade. Additionally, we compared histopathology findings from preoperative biopsies with definitive pathology results in patients who underwent radical prostatectomy (RP), aiming to determine if the SUVmax value can predict a higher risk in definitive pathology than initially indicated by the biopsy.

Prostate cancer cells commonly exhibit elevated PSMA expression, allowing for targeted PET imaging using PSMA ligands. Among these, Ga68 PSMA has shown a strong affinity for PSMA [11]. Benign prostatic epithelium also shows elevated PSMA expression, as demonstrated by immunohistochemical studies, though the expression is less intense compared to prostate cancer cells [12,13].

Overall, sufficient tracer uptake within the primary tumor is critical for effective visualization and detection of metastases via PET imaging. Utilizing 68Ga PSMA PET/CT for staging is most appropriate for prostate cancer patients who are likely to exhibit significant PET positivity in the primary tumor. Having predictive markers available at the time of initial prostate cancer diagnosis could aid in determining the suitability of primary tumors for imaging with 68Ga PSMA PET/CT.

This issue was addressed in part in recently published studies by Sachpekidis, Kopka, Eder, Hadaschik, Freitag, Pan, Haberkorn, Dimitrakopoulou-Strauss, Uprimny, Kroiss, Decristoforo, Fritz, von Guggenberg, Kendler, Scarpa, di Santo, Roig, Maffey-Steffan et al. [14,15]. In his research, Uprimny demonstrated that the intensity of tumor-related tracer uptake on 68Ga PSMA PET/CT correlates with PSA concentration and GS in newly diagnosed PC. The same upward trend in SUVmax was observed when combining Gleason scores and tumor-related tracer uptake, with higher values corresponding to increased GS/ISUP grades was found in our work.

Based on our findings, Ga68 PSMA PET/CT for primary staging of prostate cancer may be useful in tumors with a Gleason score greater than 7 or in patients with PSA levels ≥ 10 ng/mL, that is, intermediate or high-risk patients. Staging in cases with a Gleason score of 6 or 7 and PSA levels < 10 ng/mL may be futile, given the significantly lower detection rates on 68Ga PSMA PET/CT [16]. Additionally, our results showed a similar correlation between the median SUVmax values and Gleason score/ISUP grades.

### 4.1. Both Poorly Differentiated and Well-Differentiated Tumors May Not Be Detected in PSMA PET/CT

Several studies support the notion that PSMA PET/CT has reduced efficacy in detecting poorly differentiated tumors as well as well-differentiated tumors. For instance, research has shown that tumors with both low PSA concentration and Gleason scores, that is, low-risk PCa, tend to have lower PSMA uptake, which makes their detection more challenging, using this imaging modality. This is particularly problematic in patients with Gleason scores of 3 + 3 and 3 + 4, where detection rates on PSMA PET/CT are lower compared to higher scores, such as Gleason 9 [17,18]. Aykanat, Kordan, Seymen, Koseoglu, Ozkan, Esen, Tarim, Kulac, Falay, Gurses et al. found that 38 of 62 (61.2%) patients who had Gleason 8 in biopsy were downgraded after RP. Interestingly, in our work, 6 of 10 (60%) patients with GS 9–10 also were downgraded after RP. The downgrading of Gleason scores after radical prostatectomy can occur due to several reasons. One key factor is sampling error during biopsy, where the most aggressive parts of the tumor may not be captured. This leads to an overestimation of the Gleason score in the biopsy. Additionally, heterogeneity within the tumor means that different regions may show varying Gleason patterns, which the biopsy might not fully represent. These issues underscore the challenges of accurately predicting the true tumor grade based solely on biopsy [19,20].

While PSMA PET/CT is highly sensitive for detecting prostate cancer, studies have noted that it may not detect all high-grade tumors, including those with Gleason scores of 9–10. Tumor heterogeneity can result in variable PSMA expression, leading to instances where aggressive cancers exhibit low or no PSMA uptake. For example, a study demonstrated that a subset of patients with advanced prostate cancer (Gleason 9–10) showed poor detection on PSMA PET/CT due to low PSMA expression, despite the aggressiveness of their disease. These cases may require additional imaging modalities, such as FDG-PET, which can help identify tumors that are PSMA-negative but FDG-positive [21].

Additionally, smaller or poorly differentiated high-grade tumors may exhibit lower PSMA avidity, making them harder to detect. High-grade prostate cancer tumors may exhibit lower PSMA avidity due to the loss of differentiation as the tumor becomes more aggressive. This loss of differentiation can lead to reduced PSMA expression, making these tumors harder to detect on PSMA PET/CT scans. Additionally, tumor heterogeneity can result in varying levels of PSMA expression within different regions of the same tumor, further complicating detection [22]. This underlines the importance of combining PSMA PET/CT with other diagnostic tools, especially in cases of high Gleason scores.

According to Rosenzweig, Haramaty, Davidson, Lazarovich, Shvero, Haifler, Gal, Golan, Shpitzer, Hoffman et al., low Gleason scores or poorly differentiated tumors may not be detected on PSMA PET/CT [23]. In our study, one patient with a Gleason score of 10 exhibited an SUVmax of 6.14, which is comparable to values typically associated with a Gleason score of 6. However, this trend was not observed in the findings of Uprimny et al. While PSMA PET/CT generally correlates with higher Gleason scores, poorly differentiated tumors might alter PSMA expression due to changes in tumor biology or microenvironment. This could result in lower SUVmax values, as observed in our study. Differences in tumor vascularization, cellular density, or metabolic pathways could also impact PSMA uptake. These variations highlight the need for a multifactorial approach in interpreting PSMA PET/CT findings, especially in high-risk cases.

Our findings support the utility of 68Ga-PSMA PET/CT in prostate cancer staging and risk assessment. However, emerging technical approaches suggest that the integration of radiomics with clinical data could further enhance prognostic accuracy. Recent studies have demonstrated the potential of radiomics–clinical combined nomograms to predict short-term prognosis in prostate cancer. For instance, Bian, Shuying, Hong, Weifeng, Su, Xinhui, Yao, Fei, Yuan, Yaping et al. highlighted that such combined nomograms could provide valuable prognostic information beyond traditional imaging metrics like SUVmax [24]. Integrating this approach with 68Ga-PSMA PET/CT may allow for more individualized and precise patient management.

Moreover, combining PSMA-PET with standardized criteria such as PROMISE could further refine disease staging and risk stratification. Karpinski, Hüsing, Claassen, Möller, Kajüter, Oesterling, Grünwald, Umutlu, Kleesiek, Telli et al. discussed the application of PSMA-PET plus PROMISE as a powerful tool for reclassifying disease stages in a way that aligns with updated clinical risk frameworks. Utilizing these combined approaches could improve the accuracy of risk predictions, providing more reliable data for treatment decision-making [25].

In comparing our study with similar research, there are notable differences and analogies worth discussing:

Ulas Babacan, Hasbek, and Seker examined the combined use of PSMA-PET imaging and traditional staging criteria, suggesting that PSMA-PET can enhance the prognostic accuracy for prostate cancer by integrating imaging parameters with established clinical models. This aligns with our study’s findings, which also underscore the value of PSMA-PET in risk stratification, though our study focuses on SUVmax as a key parameter [26].

Bela Andela, Amthauer, Furth, Rogasch, Beck, Mehrhof, Ghadjar, Hoff, Klatte, Tahbaz et al. highlighted the potential of PSMA total lesion uptake (PSMA-TLU) as a superior predictor compared to SUVmax, especially in high-risk cases. While our study emphasizes SUVmax, Andela’s findings suggest that PSMA-TLU might offer additional prognostic insights for more aggressive forms of prostate cancer. This points to the opportunity for future research to explore multiple PSMA-PET parameters for a more nuanced understanding of tumor behavior [27].

Heetman, Paulino Pereira, Kelder, Soeterik, Wever, Lavalaye, van der Hoeven, Lam, van Melick, van den Bergh et al. explored PSMA-PET’s utility in differentiating aggressive prostate cancer, focusing on its capability to inform treatment decisions. Our study’s findings are complementary, as we demonstrate the relationship between SUVmax and tumor grade, supporting PSMA-PET’s role in treatment planning and prognosis [28].

These studies collectively reinforce the utility of PSMA-PET in prostate cancer management, though the differing focuses on specific PET parameters (e.g., SUVmax vs. PSMA-TLU) suggest that a more comprehensive approach incorporating multiple PSMA-PET metrics could enhance prognostic accuracy. Future studies may benefit from integrating these parameters for a holistic risk assessment model.

These findings indicate that while PSMA PET/CT is a powerful tool for staging prostate cancer, it is not infallible, especially in a number of both well-differentiated and poorly differentiated PCa cases.

### 4.2. Changes in Histopathological Grade after RP

Since the advent of PSA screening, there has been a substantial increase in diagnoses of low-grade, slow-growing prostate cancer [29,30]. Many of these patients are considered good candidates for active surveillance instead of aggressive treatment; as such, interventions can lead to significant side effects, including issues with urinary, sexual, and bowel functions [31,32]. However, discrepancies between Gleason scores from biopsy and those from final pathology are a well-recognized and commonly reported issue [33,34].

Based on the findings by Demirci, Kabasakal, Şahin, Akgün, Gültekin, Doğanca, Tuna, Öbek, Kiliç, Esen et al., 21% (*n* = 16) of patients who underwent radical prostatectomy shifted from a low-risk category (grade groups 1 and 2) to a high-risk category (grade groups 3, 4 and 5), while 10% (*n* = 6) of patients moved from a high-risk to a low-risk category. For patients who moved from low-risk to high-risk groups, PSMA PET images suggested that an SUVmax threshold of 9.1 would have anticipated upstaging in 10 out of 16 cases (62.5%) [35].

Consistent with previous findings, our study identified a 27.5% upgrade to grade group 2 or higher following radical prostatectomy in patients originally classified as grade group 1 based on biopsy results. Using a SUVmax cut-off of 5.64, PSMA PET demonstrated an upgrade prediction rate of 36.6%. However, the limited sample size may have contributed to lower sensitivity and specificity.

Evaluating the likelihood of clinically significant upstaging can provide important insights for initial treatment decisions and prognostic assessments. Nevertheless, this study has limitations inherent to its retrospective design. Furthermore, we did not conduct a segmental analysis of either the pathology specimens or the areas of tracer uptake observed in the 68Ga-PSMA PET/CT scans. Instead, our approach focused on the SUVmax value averaged across the prostatic lesions, which was then compared to the respective grade group. However, SUVmax has limitations, as it only represents the single most intense point of uptake and may not reflect the entire tumor’s metabolic activity. An alternative, SUVmean (SUVavr), which measures the average uptake within the lesion, could provide a more comprehensive assessment of tumor burden and heterogeneity, potentially offering a better correlation with tumor grade and clinical outcomes. A prospective study is needed to further evaluate the role of tracer uptake in predicting clinically significant cancer.

This study underscores the potential of 68Ga-PSMA PET/CT in correlating tumor uptake with Gleason scores and ISUP grades in prostate cancer patients. Our results demonstrate a significant relationship between SUVmax values and tumor grade, suggesting that a higher SUVmax is associated with more aggressive disease. The observed grade shifts between preoperative biopsy and post-radical prostatectomy pathology highlight the limitations of biopsy alone in accurate staging. These findings indicate that 68Ga-PSMA PET/CT could serve as a valuable tool in enhancing the accuracy of prostate cancer staging and guiding treatment decisions.

## 5. Conclusions

This study demonstrates that 68Ga-PSMA PET/CT imaging may serve as a valuable tool in the staging and risk stratification of prostate cancer, particularly when assessing tumor aggressiveness in relation to histopathological grading. Our findings suggest a moderately positive correlation between SUVmax and tumor grade groups, which could aid clinicians in identifying patients at higher risk and potentially guide treatment decisions. However, given the small and heterogeneous sample size, including variations in age, PSA levels, and staging, the generalizability of these results is limited. Therefore, further prospective studies with larger, more homogeneous populations are necessary to validate our findings and better understand the clinical utility of 68Ga-PSMA PET/CT in diverse patient cohorts. Such studies will be essential to establish more definitive conclusions and potentially broaden the clinical application of 68Ga-PSMA PET/CT in prostate cancer management.

## Figures and Tables

**Figure 1 cancers-16-03526-f001:**
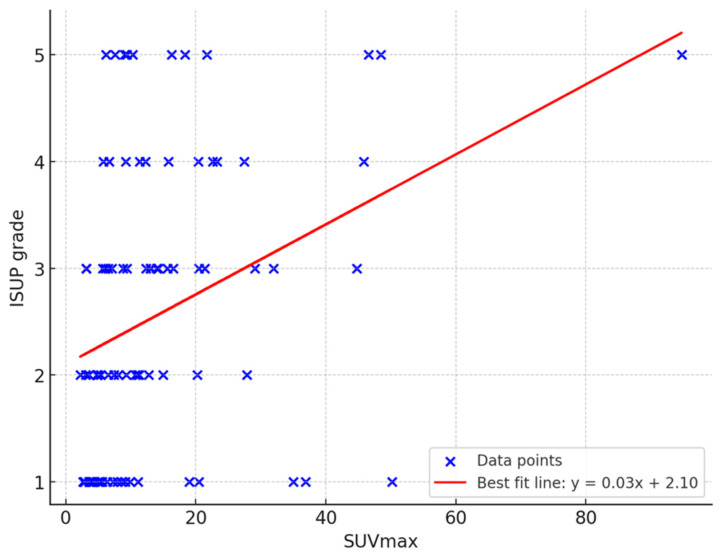
Correlation between SUVmax and tumor grade group. This scatter plot illustrates the moderately positive correlation between SUVmax and tumor grade groups in a cohort of 86 prostate cancer patients. The red best fit line indicates the relationship between SUVmax and ISUP grade, with a Pearson’s correlation coefficient of ρ = 0.34, demonstrating statistical significance (*p* < 0.001). This finding suggests that higher SUVmax values are associated with more advanced tumor grade groups.

**Figure 2 cancers-16-03526-f002:**
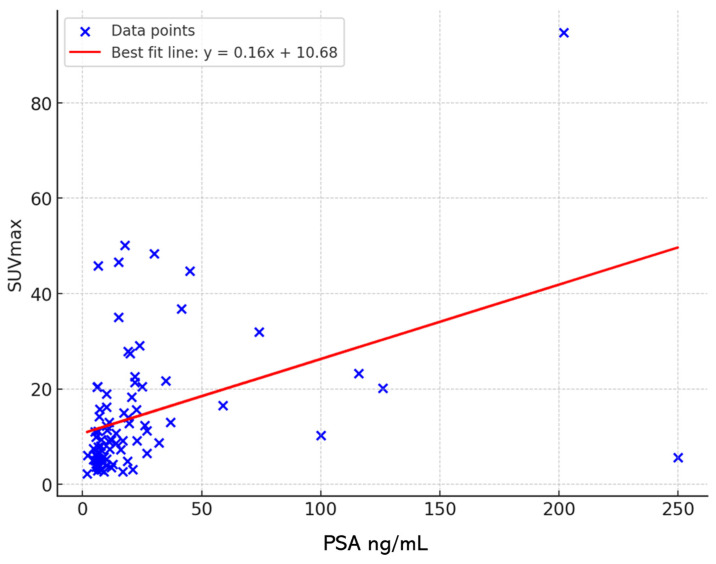
Correlation between SUVmax and PSA. This scatter plot shows a moderately positive correlation between PSA concentration levels and SUVmax values in prostate cancer patients. The best fit line, represented in red, demonstrates a Pearson’s correlation coefficient of ρ = 0.42, indicating a statistically significant relationship (*p* < 0.001). This suggests that as PSA levels increase, SUVmax values tend to rise, reflecting a potential link between higher PSA concentrations and increased tumor metabolic activity.

**Figure 3 cancers-16-03526-f003:**
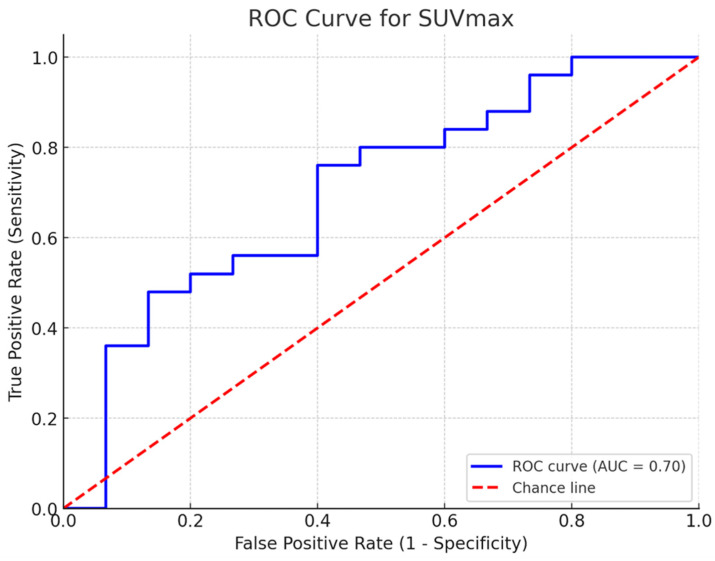
ROC Curve Analysis of SUVmax for Risk Stratification in Prostate Cancer. Following radical prostatectomy, 27.5% (*n* = 11) of patients were upstaged from low-risk (grade groups 1 + 2) to high-risk levels (grade groups 3 + 4 + 5), while 7.5% (*n* = 3) moved from high-risk to low-risk levels. An SUVmax cut-off of 5.64 was seen in 4 out of 1l patients (36.36%) who transitioned to higher-risk groups. The ROC curve analysis of SUVmax. (AUC = 0.82, 95% Cl: 0.70–0.94) demonstrated a sensitivity of 76% and a specificity of 60% at the cut-off value of 5.64, suggesting a possible but limited role in predicting risk transitions.

**Table 1 cancers-16-03526-t001:** Patients’ Characteristics.

Patients (*n*)	86
Age median (range)	68 years (42–82)
GS 6/ISUP I (*n*)	26
GS 7/ISUP II-III (*n*)	38
GS 8/ISUP IV (*n*)	11
GS 9-10/ISUP V (*n*)	11
PSA median (range)	11.3 ng/mL (1.85–250)

GS—Gleason score, *n*—number, ISUP—International Society of Urological Pathology. This table presents the demographic and clinical characteristics of 86 treatment-naive prostate cancer patients. It includes information on the age range, Gleason scores, and ISUP grades of the patients, as well as their median PSA levels. The table categorizes patients by their Gleason scores, with specific attention to the number of patients in each ISUP grade. The median age of the patients is 68 years, with a range from 42 to 82 years, and the median PSA level is 11.3 ng/mL, ranging from 1.85 to 250 ng/mL.

**Table 2 cancers-16-03526-t002:** SUVmax values and Correlation with Gleason Score and ISUP Grade.

SUVmax Values on Different GS Subgroups				
GS	N	Mean SUVmax	Median SUVmax	SD SUVmax
6	26	10.69	5.49	11.89
7	38	12.32	10.05	9.06
8	11	18.22	15.74	11.09
9	10	28.29	17.27	26.31
10	1	6.14	6.14	-
**SUVmax Values on Different ISUP Grade Subgroups**				
**ISUP**	**N**	**Mean SUVmax**	**Median SUVmax**	**SD SUVmax**
I	26	10.69	5.49	11.89
II	18	9.37	7.73	6.35
III	20	14.80	13.01	10.28
IV	11	18.22	15.74	11.09
V	11	26.21	16.25	25.87

GS—Gleason score, N—number, ISUP—International Society of Urological Pathology, SD—standard deviation.

**Table 3 cancers-16-03526-t003:** Assessment of Grade Group Shifts in Prostate Cancer: Biopsy vs. Post-Surgical Pathology.

Grade	*n* (Patients)	PSA (ng/mL)	SUVmax	*n* (Patients)	PSA (ng/mL)	SUVmax	Upgrade (*n*, %)	Downgrade (*n*, %)
1	13	9.0	4.96	5	9.0	4.0	10 (76.92%)	0 (0%)
2	8	10.0	5.075	14	8.0	5.14	1 (12.5%)	0 (0%)
3	9	19.0	12.32	16	17.0	16.15	1 (11.1%)	3 (33.3%)
4	6	6.735	17.29	2	8.45	13.21	2 (33.33%)	2 (33.3%)
5	4	17.75	14.285	3	6.57	29.08	0 (0%)	4 (100%)

The SUVmax and PSA values in this table are median values. Among 40 patients who underwent radical prostatectomy, there were 35% grade upgrades and 22.5% downgrades post-surgery, with significant changes observed in 57.5% of cases (*p* < 0.001). The table compares pre-operative PSA levels and SUVmax values with post-operative findings for each grade group. It illustrates how changes in these parameters correspond to upgrades or downgrades in tumor grade, highlighting the discrepancy between biopsy and definitive pathology results, and emphasizing the need for accurate pre-operative assessments.

## Data Availability

The data that support the findings of this study are available from the corresponding author(s) upon reasonable request.

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
