# Peer review of "Enhancing Prostate Cancer Staging: Association of 68Ga-PSMA PET/CT Imaging with Histopathological Grading in Treatment-Naive Patients"

_cancers, 2024, doi:10.3390/cancers16203526_

Round 1
Reviewer 1 Report
Comments and Suggestions for Authors
The authors in this manuscript presented 86 prostate cancer patient data administrated with 68Ga-PSMA for PET/CT Imaging study. The authors predicted sensitivity 76% and specificity 60% in upstaging patients, how both parameters were measured, pls explain.
Results are well presented, Discussion was also done using good references
The overall quality of study is good and can be considered for publication.
Reviewer 2 Report
Comments and Suggestions for Authors
This retrospective study analysed the correlation 68GaPSMA uptake in prostate cancer patients and histopathological grading with Gleason Score and ISUP grade. Also, it compared preoperative pathology and postoperative pathology. The design, methodology and results are adequately performed and presented.
Comments:
1. Limitations of the study: The limitations sections should include the fact that the population studied was small and heterogeneous in many aspects: age, PSA values, staging, etc. This undoubtedly limits the grade of generalizability of the results. Because of this, in the Conclusion section it should be underlined that there is a need for further prospective studies with homogeneous populations.
2. Conclusions: Given the limitations of the study I suggest rephrasing Line 262 "PET/CT can serve as a valuable tool in e " for "PET/CT could serve as a valuable tool in e "
3. Introduction: Line 37. Prostate cancer is one of the tumours in which it is key to work in a multidisciplinary team in order to obtain the best possible results for patients. I suggest incorporating this and referring to the following international guidelines: Brausi M, Hoskin P, Andritsch E, Banks I, Beishon M, Boyle H, Colecchia M, Delgado-Bolton R, Höckel M, Leonard K, Lövey J, Maroto P, Mastris K, Medeiros R, Naredi P, Oyen R, de Reijke T, Selby P, Saarto T, Valdagni R, Costa A, Poortmans P. ECCO Essential Requirements for Quality Cancer Care: Prostate cancer. Crit Rev Oncol Hematol. 2020 Apr;148:102861. doi: 10.1016/j.critrevonc.2019.102861. Epub 2020 Jan 7. PMID: 32151466.
4. Discussion-Line 181. In poorly differentiated prostate cancer tumours FDG PET/CT can be useful, as there is evidence of FDG uptake in dedifferentiated tumours. In complex situations, the combination of multitracer studies with PSMA and FDG can provide information on the tumor biology, as is frequently done in NET.
Reviewer 3 Report
Comments and Suggestions for Authors
In this study, you correlated the SUVmax value observed in the prostatic bed with GS, ISUP grades and PSA values. The goal was to validate a prognostic value of PSMA PET scan improving the biopsy only results.
The paper is correct from the methodological point of view.and the results results confirm a potential role of PSMA-PET as previously reported in literature, although SUVmax parameter cannot be considered a strong prognostic indicator.
*. I suggest to integrate in your discussion further technical approaches as the radiomics-clinical combined nomograms able to predict short-term prognosis (Bian S et al. Abdom Radiol (NY). 2024 ;49:3747) or combining PSMA-PET plus PROMISE to redefine disease stage and risk Karpinski MJ. Lancet Oncol. 2024;25:1188).
*. Differences and analogies with similar studies ( Ulas Babacan O et al. Curr Oncol. 2024;31:5307 / Bela Andela S et al. Radiat Oncol. 2024;19:97 / Heetman JG et al. Prostate. 2024;84:1025) are worth comparing.
* Many references are reported with different styles, some of them are incomplete.
* row 189 typo error: "who what Gleason"
Reviewer 4 Report
Comments and Suggestions for Authors
The cancers-3251399 is an interesting topic. The authors investigated the value of 68Ga-PSMA PET/CT for the initial staging of PCa patients with this retrospective study. The overall quality of the manuscript is good.
Abstract
This section is complete and summarizes effectively the topic of the manuscript.
Introduction
The introduction is concise. It successfully provides a solid foundation for the study, which makes the comprehension easier. The relevant background regarding the physiology of 68Ga-PSMA PET/CT is presented. In addition, the research question is clearly stated in this section.
Materials and methods
The inclusion criteria are mentioned in this section. The image acquisition and analysis are properly analyzed.
Results
The results are presented in an extensive way, providing a comprehensive overview of the findings of this study. The figures and tables are necessary for the completion of the authors’ work. The correlation between greater 68Ga-PSMA uptake in patients of higher stage is clearly presented.
Discussion
This section is complete and includes updated data. The limitations of the study are thoroughly presented, and the value of a prospective study is correctly underlined by the authors.
Conclusion
The conclusion is complete and represents the work that the authors did in this study.
